# Has Biomimicry in Architecture Arrived in France? Diversity of Challenges and Opportunities for a Paradigm Shift

**DOI:** 10.3390/biomimetics7040212

**Published:** 2022-11-23

**Authors:** Estelle Cruz, Eduardo Blanco, Fabienne Aujard, Kalina Raskin

**Affiliations:** 1CEEBIOS, French Network in Biomimetics, 75004 Paris, France; 2MECADEV UMR CNRS 7179-National Museum of Natural History of Paris, 91800 Brunoy, France; 3CESCO UMR 7204-National Museum of Natural History of Paris, 75005 Paris, France

**Keywords:** biomimicry, France, paradigm shift, public policies, architecture, urban design, arts, education, research, urbanism, demonstrators

## Abstract

Biomimicry is a growing field of developing environmental innovations for materials, facade systems, buildings, and urban planning. In France, we observe an extensive diversity of initiatives in biomimicry for the development of regenerative cities. These initiatives blossom in a large range of areas, from education to urban policies, to achieve a major environmental, social and economic paradigm shift. To provide a comprehensive understanding of this development at the national scale, this paper presents and discusses the diversity of the developed initiatives over the last 10 years in six main fields-education, urban policies, fundamental and applied research, design demonstrators, arts, and communication. This research is an opportunistic study based on the analysis of these initiatives enriched by the feedback of the stakeholders collected by the authors working in the field of biomimicry over the last seven years. We identify that biomimicry in France has mainly extended through individual initiatives of teachers, territorial authorities, architectural studios, or researchers rather than through the support of public policies. Putting into perspective developments in biomimicry by other countries, this cross-discipline analysis provides recommendations for the extensive development of regenerative architecture and urbanism at the national scale.

## 1. Introduction

### 1.1. Biomimicry for a Paradigm Shift

‘Biomimetics’, ‘biomimicry’, or ‘bioinspiration’, defined as the transfer of strategies from biology to technology, is a growing research area between Life Sciences and Design Sciences [1,2]. ‘Biomimicry’ refers to a ‘*philosophy and interdisciplinary design approaches taking nature as a model to meet the challenges of sustainable development,*’ while biomimetics refers to an “*interdisciplinary cooperation of biology and technology or other fields of innovation with the goal of solving practical problems through the functional analysis of biological systems, their abstraction into models, and the transfer into and application of these models to the solution*” according to ISO 2015:18458 [2]. The main significant difference between ‘biomimetics’ and ‘biomimicry’ is that the approach referring to the latter tends to be specifically focused on developing sustainable solutions; the former does not have to fit that requirement. Like ‘biomimetics’, ‘bioinspiration’, defined as a ‘*creative approach based on the observation of biological systems’,* [2] does not have to meet sustainable goals.

In built environment disciplines, biomimicry is an emerging approach to develop environmental innovations for materials, facade systems, buildings, and, more recently, urbanism [3,4]. Beyond famous and inspiring success stories of biomimicry, such as the Eastgate building in Harare, the approach of biomimicry has great potential to develop urban and architectural regenerative projects with a positive impact on social, economic, and ecological systems [4,5,6].

### 1.2. The International Context

International interest in biomimetic architecture has increased since the seventies within a global context of energy transition due to the oil crisis in 1973 and 1979, environmental awareness, and the development of biomimetic success stories. As proposed by Michael Pawlyn, the development of biomimicry in architecture can be understood through the hype cycle model, which traces the evolution of technological innovations as they pass through the five following successive stages presented in Figure 1: (i) innovation trigger, (ii) peak of inflated expectations, (iii) trough of disillusionment, (iv) slope of enlightenment, (v) and plateau of productivity [7,8].

Iconic biomimetic success stories, such as the building of the West German Pavilion inspired by insects’ light habitat in 1967, the discovery of the self-cleaning hydrophobic surfaces relying on the Lotus effect in 1976, or the development of the self-ventilation system of the Eastgate building in 1996, respond to the phase (ii) peak of inflated expectations since they were highly publicized by international media [9]. However, most of these biomimetics projects emerged from a biology push approach, in which a biological property is observed and then transferred to solve a technical problem [2,10,11]. In fact, methodological obstacles are primarily related to biological data, where access, understanding, and selection remain the main challenges [11,12]. Despite the growing interest of the public, the media, and academic and industrial actors, these barriers resulted in the ‘phase of disillusionment’ since today, products’ or buildings’ development mostly follow the biomimetic technology pull design process, as defined by [2]. At that time, methods and tools were not mature yet to facilitate this process, and biomimetic designs emerged from the initiative of designers, architects, or researchers. This awareness of the issues related to biomimetic application in architecture corresponds to the disillusion phase (iii).

Over the next twenty years, the development of more that 43 methods and tools has helped to support the development of biomimetics in all fields [13,14]. Particularly, the ISO 2015:18458 Standard, which allowed for an international formalization of the semantics associated with biological knowledge transfer, main steps, and defined criteria to classify biobased approaches [2]. Likewise, some tools were specifically proposed to support architects, designers, and urbanists in applying biomimetics, such as the tools Genius of Biome [15], BioGen [16], and ESA (Ecosystemic Services Analysis) [17]. In parallel, the development of incubators, research centres, and funding programs dedicated to the development of biomimetics, including architectural designs—e.g., the Collaborative Research Centre SFB-TRR 141 in Germany (Stuttgart – Tübingen – Freiburg Universities) [18], the FIT (Freiburg Center for Interactive Materials and Bioinspired Technologies (University of Freiburg)) [19], and the Bio-inspired Material National Centre of Competence in Research in Switzerland (Fribourg University) [20]—have highly contributed to overcoming the disillusion phases (iii), reaching the current slope of enlightenment (iv). 

**Figure 1 biomimetics-07-00212-f001:**
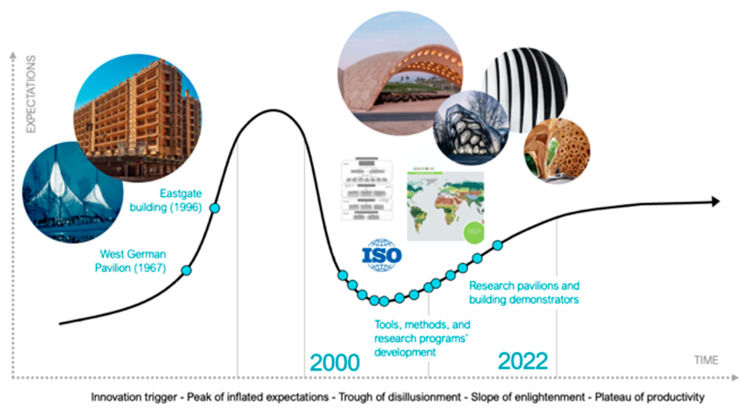
The development of biomimicry in architecture and urbanism understood through the hype cycle model. Content and information adapted from various sources: [8,14,17,21,22,23,24,25]. Credit: CC-BY-SA Estelle Cruz.

### 1.3. The French Context

In France, biomimicry is a growing field of innovation in different topics [26,27,28]. In 2011, the French translation of the book ‘*Biomimicry: Innovation Inspired by Nature*’ by Janine Benyus constituted a turning point in reaching a wider audience [1]. At the same time, the francophone NGO ‘Biomimicry Europa’ was founded by a group of French and Belgian designers, architects, and biologists to promote the approach through various actions such as exhibitions, conferences, and involvement in research projects [29]. They were mostly carried out by volunteers until the creation of the Ceebios in 2014-a not-for-profit French network and center of expertise on biomimicry [30].

In 2022, Ceebios employed 30 full-time workers, counted more than 400 partners, and conducted a wide range of activities to increase the development of biomimicry in France. The Ceebios’ bioinspired habitat section focuses on (i) the facilitation of working groups for academics and industrial partners to increase collaboration [30], (ii) the development of training for professionals, postgraduate, and undergraduate students in partnership with French Institutions [31,32], (iii) assisting building constructors, architecture studios and French policies for the development of bio-inspired urban planning, buildings or facades [33,34] and, (iv) developing research on bio-inspired architecture and urban design [35,36]. Among all these actions, the authors observe that the development of biomimetic building demonstrators has been the main initiative, as recently documented by Blanco et al. [28], and Chayaamor-Heil et al. [37].

### 1.4. Research Objectives

Beyond the development of sustainable and innovative designs at the architectural scale, the philosophy of biomimicry can be encompassed as a global society’s response to a paradigm shift and the development of regenerative societies [1,38,39,40]. However, long-term environmental, social, and economic shifts must go through several steps, involving a wide range of stakeholders to be embedded at all layers of societies.

To provide a comprehensive understanding of the development of biomimicry for a long-term paradigm shift, this paper presents and discusses the diversity of the developed initiatives over the last 10 years in six main fields-education, urban policies, fundamental and applied research, design demonstrators, arts, and communication. This research is based on the analysis of these initiatives enriched by the feedback of the stakeholders of the French building sector collected by the authors working in the field of biomimicry in architecture and urbanism over seven years. This study aims to understand the biomimicry development processes on a country scale, using France and the subject development in its territory as a case study.

## 2. Methodology 

### 2.1. Data Collection

As an opportunistic study, this research did not produce data but collected and analysed data from scientific papers, media, and online documents published by architects, teachers, territorial authorities, etc. The collected data are sorted into 6 main categories described in Section 2.2. Data classification. Each category is introduced by a table that presents the selected initiatives. Rather than giving an exhaustive mapping of initiatives for the development of regenerative cities, this research outlines the diversity of actions as extensive. Each category highlights several initiatives or projects which meet the two following criteria:They refer to the definitions of either *bioinspiration*, *biomimicry* or *biomimetics* according to [2]. They promote different understandings of sustainability, comprehension, and abstraction of living systems; however, this is not evaluated in this study.They are both carried out by French organizations—such as universities, architecture studios, etc.—and undertaken in France. International cases of study are not detailed in this paper.

### 2.2. Data Classification

The undertaken actions for the development of regenerative architecture and urbanism through the biomimicry approach have been sorted into six categories: **Design of demonstrators:** Development of prototypes of bioinspired materials, facades, or buildings;**Fundamental and applied research:** Development of tools and methods, and prototypes through collaborative programs, doctoral and post-doctoral research;**Urban policies:** Development of public bids which require the integration of biomimicry;**Education:** Development of training in bioinspired architecture, civil engineering, and urbanism for the undergraduate, postgraduate, and professional public;**Art and Inspiration:** Development of art and inspiration based on the principles of biomimicry for futures cities and buildings;**Communication:** Awareness-raising actions for the public, such as conferences, exhibitions, podcasts, and documentary.

### 2.3. Data Analysis

Results were analysed using a qualitative perspective and enriched by the feedback of stakeholders of the building sector collected by the authors working in the field of biomimicry over the last seven years. Each section first provides the context, then presents the French biomimicry projects, and discusses within an opportunity section that aims to explore lessons and barriers to further development.

### 2.4. Methodological Limits

This research is an opportunistic study and does not include a quantitative analysis of the data. Further developments must assess the impact of these initiatives for a paradigm shift. 

## 3. Opportunities to Integrate Biomimetics in Architecture and Urban Planning

### 3.1. Design Demonstrators

In this section, we looked at design demonstrators such as buildings or façade systems built over the last 10 years and referred to as bioinspired. They were designed by French architecture studios, which have also designed worldwide biomimetics projects, as reviewed by Blanco et al. [28]. The diversity of demonstrators is presented in Table 1.

#### 3.1.1. Design Demonstrators in Biomimicry

New building systems, such as building materials, façades, or roof systems, must undergo a technical assessment process before being sold on the French market (ATEc) or used for a specific innovative building project (Experimental Technical Assessment—ATEx). These assessments are delivered by a public institution, the CCFAT—French Commission for Technical Advice, represented by the CSTB—French Scientific and Technical Center for Building [46]. The CSTB is a public industrial and commercial company (EPIC) in charge of the technical evaluation of these novel building systems [47]. An ATEx is usually undertaken before an ATEc application. For technical advice, applicants must provide scientific and technical validity supported by evidence. This certification time varies from a few months to several years.

#### 3.1.2. Design Demonstrators in Biomimicry

There is a great diversity of design demonstrators in biomimicry due to the variety of their type, year of construction, and level of innovation. As demonstrated by Cruz and Hubert et al. [48], there are two main ways to integrate biomimicry for the development of a novel building: (i) an innovative combination of existing building systems as carried out for the construction of the D2 tower or the Spiral House (Figure 2b,d) or (ii) the development of radical and incremental innovations, implemented through novel and uncommon manufacturing techniques as the CIRC façade or the Biolum’Reef project (Figure 2a,c,e,f). The biomimetics projects that resulted from the innovative combination of existing building systems do not require additional technical evaluation, such as ATEc or ATEx, while radical innovations in biomimicry must attend these evaluations. 

Following the biomimicry projects’ classification proposed by Pedersen Zari [6], these French projects can also be sorted in two main categories: projects’ that are inspired by a specific biological model, such as the CIRC inspired the opening and closing systems of flowers, or ecosystemic projects that combine several environmental principles such as carbon reduction, circular economy, nature-based solutions or biomimicry. Few projects combine both approaches, except the project Ecotone in Arcueil. The design method of this mix-use building combined several tools and environmental requirements derived from several biomimicry approaches. For instance, the building façade integrates principles abstracted from adaptive living systems, such as the human skin for thermal regulation and pinecones’ closing movement abstracted for the light regulation of the building façade. The design teams also integrated several ecosystemic services provided by the building [41,49]. 

Despite the diversity of demonstrators, none of them can be considered as regenerative despite some trying to meet the requirement of the Living Building Challenge certification [50] e.g., Ecotone, and the project ‘Osez Josephine’, or have applied a biomimetic method.

#### 3.1.3. Opportunities Design Demonstrators in Biomimicry

Based on the analysis of these six building demonstrators, the feedback of stakeholders and our field observations, we propose some recommendations. We believe that for better development of biomimicry demonstrators, further research must provide an environmental evaluation of existing bioinspired buildings and evaluation grids to help the design teams evaluate their projects during the design process.

However, these case studies provide cases of studies of implementing biomimicry to building demonstrators. Next, biomimetics and regenerative projects must go beyond the existing in terms of sustainability and design methods. In addition, the development of these biomimetic projects is limited compared to the number of new buildings constructed each year in France.

### 3.2. Fundamental and Applied Research

This section analyses the French research programs carried out over the last ten years in biomimicry for the built environment. The diversity of research in that field is sorted into two main categories: (i) research programs for building systems’ development and (ii) research programs for methodological development. Results are presented in Table 2.

#### 3.2.1. Research in the French Context

Short-term research has always been part of architectural and urban planning practices [3]. Architects carry out literature and on-site analysis, interviews, drawing and design research for every single project as each differs from the previous one. Then, the design team draws proposals to address the multi-criteria requirements the project must meet. However, very few French architecture studios have conducted research in the academic sense. The fundamental differences between these fields can be highlighted by the three following striking examples: the building design timescale varies from weeks to months, whereas academic research is spread over years. French architecture schools are dependent on the Ministry of Culture since 1995-rather than Education or Research, and the doctorate in architecture was only created in 2005 when the studies of architecture in France were reformed into a DMI system (Doctorate, Master, License).

#### 3.2.2. French Research Programs in Biomimicry and the Building Sector

Research programs for system development. In this context, few architecture studios have carried out long-term, and applied research beyond the construction of a single building. Focusing on biomimicry, some research remains an exception, as presented in Table 2. For instance the SymBiO_2_ program by XTU Architects has allowed for the development of a photobioreactors façade system inspired by an autotroph marine slug (*Elysia chlorotica*), in partnership with the laboratory GEPEA of the University of Nantes [51]. Likewise, Tangram Lab developed bioluminescent facades systems in partnership with the Mediterranean Institute of Oceanography of Marseille [53,54]. The architecture studio ArtBuild developed an adaptive shading system that mimic passive mechanisms observed in plants. This system is currently tested within the building façade of the CIRC building in Lyon in France [23,55]. 

These breakthrough systems are being developed alongside the main activities of architecture studios, from the concept to the building technical evaluation. The business risk is high for architecture studios regarding their revenues. In France, only 12% of architecture studios have a turnover of more than 500,000 euros. More than 70% of architects worked alone or with a single employed [57]. As observed by the authors, most architects promote the use of mature biomimetics building systems since using non-mature building systems requires additional assessment steps regarding building code requirements [48].

Research programs for methodology development. As outlined by several studies, the methodological aspects—research, access, transfer of biological knowledge to architecture or urbanism, and then evaluation of the bioinspired system—remain one of the main challenges [11,12]. For this purpose, six doctoral theses focused on methodological aspects in biomimicry to enhance its development over the last ten years. Three of them focused on the design process [58] and proposed frameworks to clarify the key roles of biologists [59] and designers [60]. Two others developed a tool for the design of multi-functional facades inspired by biological envelopes [35] and supported by the development of a façade prototype [56]. The last focused on developing guidelines for applying biomimicry in the urban environment [36]. Likewise, the two-year project BiomimArchD aims to develop a design assistance tool based on an ontology of biomimetic architecture knowledge [61]. The tool is designed for architects to aid in knowledge transfer. The company Saint Gobain Isover, which develops building insulation solutions, carried out a comparative review of animals to find innovative thermal regulation processes for building. This prospective research was undertaken by biologists [62]. 

In addition, research efforts have focused on the development of interdisciplinarity programs such as the applied research BiOMIg on bioinspired materials for cross-industry sustainable applications, including building materials. This multi-disciplinary consortium gathers different organizations such as academics, industrial, public and private research [63].

#### 3.2.3. Opportunities to Enhance Research in Biomimicry

Based on the analysis of these research developments, the feedback of stakeholders and on our field observation, we propose some recommendations. We believe that the support of public policies will highly help a better development of biomimicry in architecture and urbanism since most of the initiatives arise from the individual efforts of researchers or architecture studios and are mostly self-funded. For this purpose, public policies must support research in biomimicry through interdisciplinarity funding programs such as the MITI-Mission for Transversal and Interdisciplinary Initiatives of the CNRS–the French National Center for Scientific Research—created in 2017 [64].

### 3.3. Urban Policies

In this section, we looked at the French urban and building tenders that request ‘biomimicry’. Table 3 outlines the five main calls published over the last 10 years that required a bioinspired, biomimetic and/or regenerative building or urban planning proposal [65].

#### 3.3.1. Urban and Building Bids in FRANCE

In France, the code governing public works contracts-Code des Marchés Publics-defines the rules to award public building contracts. Public contracts for architectural competitions are awarded in two selection phases: (i) the offer phase and (ii) the application phase. These two phases are evaluated by territorial authorities, which define the program and the specific environmental requirements. Over the last twenty years, several novel environmental requirements were requested in architectural and urban competition, such as the integration of the principles of the circular economy, biophilia, cradle-to-cradle, natural steps, etc.

#### 3.3.2. Biomimicry in Urban and Building Tenders

In that context, four territorial authorities have supported the integration of biomimicry in their urban and building public tenders. In 2018, the Nouvelle-Aquitaine region put out to tender for the construction of the Marine Biomimetic Centre of Biarritz. This architectural design competition has integrated environment building requirements based on biomimicry specifications and the Living Building Challenge framework and mindset (without aiming the certification) in 2018. This 5.800 square meters open-innovation site will host research teams, students, and international companies (Figure 3) [67]. Two years later, the Nouvelle-Aquitaine region also put out a tender for constructing a theatre hall of 120 square meters [68]. Aligned with the environmental requirements of the Marine Biomimetic Centre of Biarritz, the theatre must result from a biomimetic design process [66]. Likewise, the architectural competition for the conservatory of the city of Senlis integrates biomimicry requirements at the request of the Hauts-de-France region [69]. In 2021, the town-planner SLP 217 published the first call for tender to develop a bioinspired programme and urban planning for 35 hectares of a mixed-use business park [70].

#### 3.3.3. Opportunities to Enhance Biomimicry in Bids

Based on the analysis of these four bids, the feedback of stakeholders and our observations, we propose some recommendations. They are derived from the sourced document-*Projets urbains bio-inspirés: un état des lieux des projets français, Biomim’City Lab, Paris, France,* which result from several workshops carried out with the members of the Biomim’City Lab, a French collaborative workgroup of biomimicry practitioners created in 2019 [49,71].

Dozens of calls for tender are published every month for urban development or the construction of public buildings, while four calls in biomimicry have been published since 2018. Beyond these four initiatives supported by territorial authorities, we believe that public policies must support the integration of environmental requirements in biomimicry. For instance, integrating strong environmental demands within public urban documents is key to enhancing regenerative urban planning and design development. The publication of the document ‘*Drawing inspiration from life for the ecological transition of buildings*’ in 2022, by the public organisation ADEME—the French Environment and Energy Control Agency—is an example of the positioning of public policies needed to develop the approach [72].In addition, territorial authorities and design teams need to provide a qualitative and quantitative evaluation of urban and architectural projects that refer to biomimicry. At that time, only the Marine Biomimetic Centre of Biarritz incorporated recommendations for implementing biomimicry based on ecosystem services and the Living Building Challenge label [73].

### 3.4. Education

In this section, we looked at the undergraduate, postgraduate, and professional courses in biomimicry in architecture or urbanism. Training is sorted into two main categories in Table 4: (i) short-term courses for undergraduate students and (ii) long-term courses for postgraduate students and professionals. The literature also counts a large range of introductive courses in biomimicry which is completed in less than two hours. These courses are not considered in that section.

#### 3.4.1. Education in Urban Planning and Architecture in France

The degree programme in architecture is a 5-year undergraduate programme. The training is divided into two parts: the teaching of architectural design, which is profession-oriented, and short-term courses that underpin the architectural practice in the form of lectures, tutorials, or seminars. These courses enrich the practice considered the core of teaching [77]. After completing their degree, architects can pursue further training through short professional courses (one to several days). These are mostly paid for and financed by public funds such as the CPF, French Professional Training Credit, or the employer in order to acquire a novel skill necessary for the current job [78].

#### 3.4.2. Teaching Biomimicry in Architecture and Urbanism

Short-term courses. There is a wide range of training in French schools and Universities for the development of bioinspired buildings, urbanisms or building materials [79]. Since biomimicry was mostly taught through introductive lectures in early 2010, current curriculums now offer short-term courses to learn biomimicry through a narrow-defined scope, such as bioinspired envelopes, for instance, at the National School of Architecture of Grenoble and Paris Val de Seine [32], biomimicry and urbanisms at the University of Toulouse [74] or digital architectural design and biomimicry at the National School of Architecture Paris la Villette. As outlined in Table 4, this student training is mostly optional courses embedded in larger programmes, open to a small number of students, and most of the time taught by only one teacher.

Long-term courses. In 2022, none of the French university curriculums has placed biomimicry as a central topic to obtain a master’s degree in architecture, urbanism, or civil engineering. The master’s programmes NID—Nature Inspired Design—of the design school of ENSCI in Paris and the BIM—Master of Bioinspired Material—of the University of Pau are the only comparable examples in France since biomimicry is embodied throughout all training [80]. As a response to the lack of training in biomimicry in urbanism and architecture, the first French professional training for stakeholders of the built environment was created in 2021 [30]. This two-day training course is based on the analysis of case studies, results from applied research and scientific publications.

#### 3.4.3. Opportunities to Enhance Biomimicry in Education

Based on the analysis of these curricula, the feedback of stakeholders and on our observations, we propose some recommendations:The teaching of biomimicry must be permanently integrated into undergraduate curricula. Teaching must not only rely on the initiative of one teacher. A good example would be the two-semester curriculum in biomimetics for undergraduate students taught by the ITECH program of the University of Stuttgart [81,82];French architectural schools must offer master’s degrees in architecture, urbanism or civil engineering, which place biomimicry as a central field. At the international level, the One Studio program offers a one-year post-professional Master’s training for the development of bio-inspired construction at the University of California [83].

### 3.5. Arts and Inspiration 

#### 3.5.1. Arts, Architecture, and Civil Engineering in France 

Architecture has always been strongly linked to civil engineering at the international level. However, in the 19th century, the French government deeply changed the educational curriculum by splitting architecture and civil engineering. Architecture was then considered as one of the four disciplines of the Fine Arts schools, along with engraving, sculpture, and painting. The Fine art School of Paris ensured the teaching of architecture until the creation of several architecture schools in 1968. Although both schools teach technical and artistic curricula, arts, architecture, and civil engineering are still considered distinct fields in the professional sectors [84]. As a result, collaborations between architects, artists, and civil engineers remain difficult. As a response, several French architectural firms have started to employ both engineers and architects. Likewise, double degrees in both architecture and civil engineering were created in 1980 to reduce the gap between disciplines [85]. 

#### 3.5.2. Biomimicry from Art Nouveau to Futurism Cities

In this context, biomimicry is both considered by French architects as a novel source of inspiration for sustainable design in line with solar architecture, developed in the mid-seventies, and as a novel source of nature-based design, in line with the Viennese industrial Art Nouveau architecture. In contrast, most French engineers and biologists considered biomimicry only as a source of innovative solutions for sustainable and regenerative design. As observed by the authors, French scientists do not consider biomimicry as a novel source of inspiration for aesthetic aspects since they fear that aesthetics will prevail over sustainability. These projects only inspired by living systems for aesthetic considerations are often called biomim’washing. 

Despite these debates within the profession, biomimicry highly stimulates the imagination of architects and artists. For instance, the belgian architect Luc Schuiten designed a series of drawings call the ‘Vegetal Cities’, which embodied the concept of biomimicry [86]. Metropolises are imagined as vast orchards, which created symbiotic interactions between humans, fauna, flora, and all the living systems. The concept of archiborescence developed by the architect was applied to imaginary cities and existing metropolises, such as the cities of Lyon and Strasbourg (Figure 4), Bruxelles, Sao Paulo, Venise, Shanghaï, etc. [86]. This work is in line with the ongoing developments of nature-based solutions, regenerative and biophilic cities. The French architecture studio of Vincent Callebaut also developed futurist utopia of bioinspired cities and buildings. His work combined the concepts of ecosystem services, high technologies, and biophilia [87].

#### 3.5.3. Opportunities to Enhance Biomimicry in Arts

Based on the analysis of the context of arts, sciences, and biomimicry in France, the feedback of stakeholders and on our observations, we propose some recommendations:

Beyond the debates between disciplines, the development of urban utopia is necessary to help the building sector to imagine sustainable and restorative futures. Since imagining an innovation from the study of a biological system is a long process, the graphics, drawings, and paintings that embodied an interpretation of biomimicry are key for its development. These representations also contribute to reducing the gap between the concepts of biomimicry for regenerative cities and their implementation. 

There is also a need for a diversity of representations and interpretations of biomimicry for restorative cities since only these two representations were designed at the international scale. 

### 3.6. Communication 

#### 3.6.1. The Development of Novel Approach

As presented in Section 1.2, the international context, the development of a novel approach, can be understood through the five main steps of the Hype cycle. At all stages, the media play a key role in its development, especially in the steps (ii) peak of inflated expectations and (iii) trough of disillusionment, where the technology is highly publicized. 

#### 3.6.2. Communicating on Biomimicry for Regenerative Cities

A wide diversity of initiatives has been carried out over the last ten years in the field of biomimicry for architecture and urbanism. For instance, the French translation of the book ‘Biomimicry in Architecture’ by Michael Pawlyn, in 2019, allowed for a better understanding of the potential of biomimicry for buildings. Likewise, several technical reports [34,65,72,88], newspaper articles in the French media, documentaries, such as the series Nature = Future [89], or podcasts [90], have significantly advanced knowledge in that field. 

We also count several events throughout the country, on sustainable cities and buildings, which include panel discussions on biomimicry. One of them, the Biomim’Expo created in 2015 by NewCorp Conseil, is an annual event only dedicated to biomimicry [91]. This two-day event gathers several stakeholders in the building sectors, biologists, designers, and politicians to present and discuss advances in biomimicry for cities. 

#### 3.6.3. Opportunities to Enhance Biomimicry

The authors strongly believe that communicative efforts through media, documentaries, podcasts, and events must continue to reach different audiences from professional to civil society. 

## 4. Conclusions

This paper presented a diversity of initiatives carried out in France to enhance biomimicry within architectural and urban design and planning practices. Depending on the project context and the involved actors, the development of biomimicry can be undertaken in different ways and at different design stages. 

The initiatives in education, fundamental and applied research, urban policies, art and inspiration, communication, and building demonstrators’ development must be pursued in parallel to reinforce each other. 

Beyond individual initiatives, we believe that public policies must support the systematic integration of biomimicry for the development of regenerative cities and buildings.

## Figures and Tables

**Figure 2 biomimetics-07-00212-f002:**
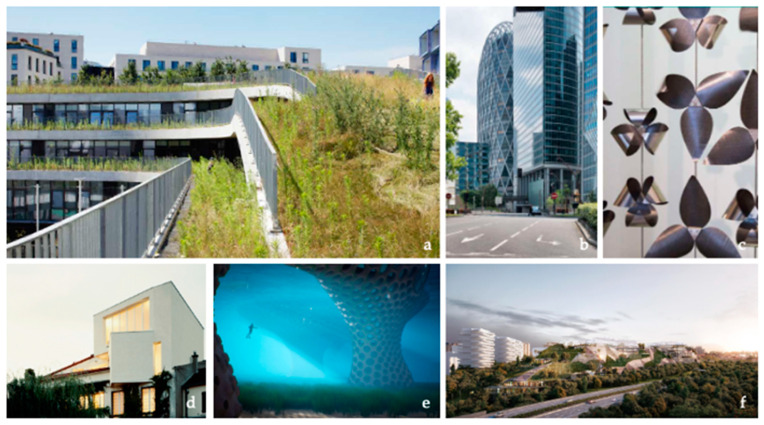
Built bioinspired buildings located in France (**a**) Ecole des sciences et de la biodiversité, Boulogne-Billancourt, 2015 © ChartierDalix – Takuji Shimmura, (**b**) D2 tower © Pierre Elie de Pibrac, (**c**) © ArtBuild, (**d**) © In Situ Architecture, (**e**) © Rougerie+Tangram, (**f**) © Tryptique Architecture.

**Figure 3 biomimetics-07-00212-f003:**
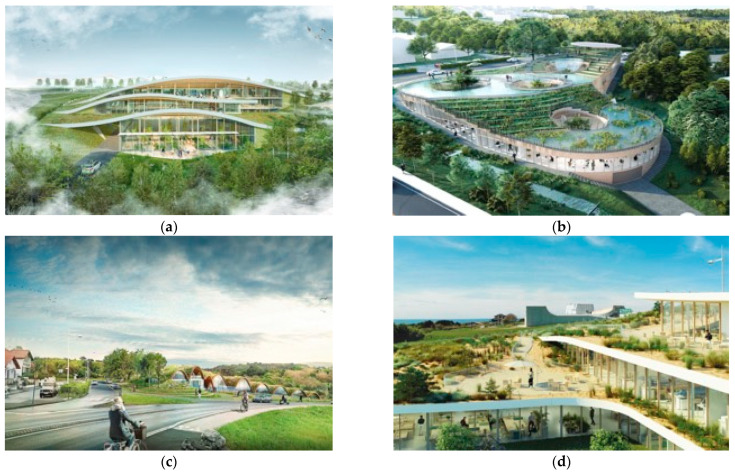
Proposed projects for the Marine Biomimetic Centre of Biarritz, 2019 (**a**) winning project © Arotcharen architecture studio, (**b**) Estran © Bechu & Partners, (**c**) © Tangram & Tangram Lab, (**d**) © ChartierDalix.

**Figure 4 biomimetics-07-00212-f004:**
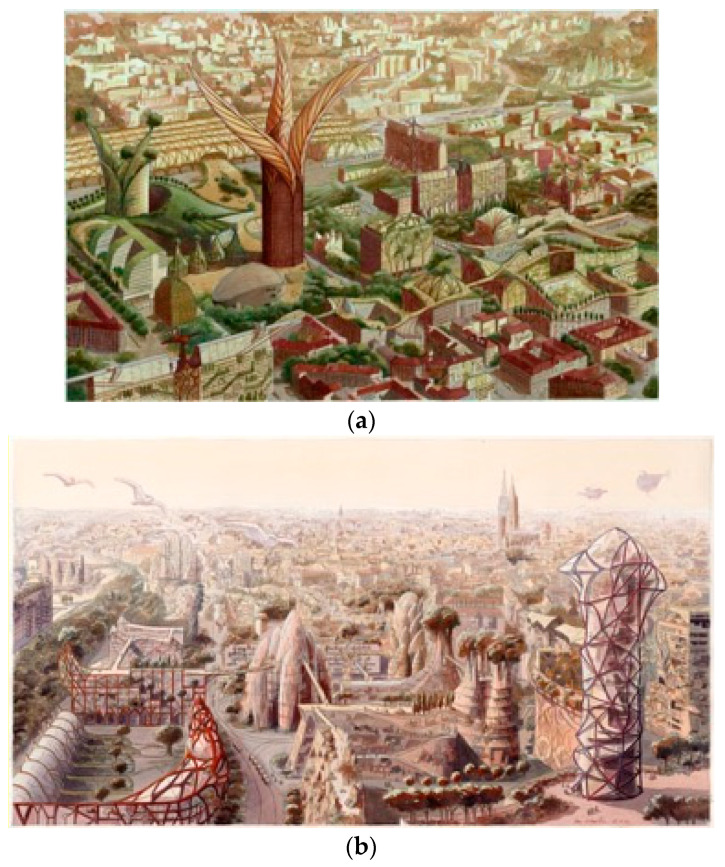
Vegetal cities, © Luc Schuiten. (**a**) Lyon in 2100, (**b**) Strasbourg in 2100.

**Table 1 biomimetics-07-00212-t001:** Examples of French built projects referred as bioinspired and located in France. The detailed of the projects can be founded in the source document ‘*Projets urbains bio-inspirés: un état des lieux des projets français*’, Biomim’ City Lab & Ceebios, 2021, Paris, France’ [34].

Project Name	Description	Status
Ecotone, Arcueil (France)	Mixed-use building designed by the Tryptique, Oxxo, and Parc Architects [41]	Under construction
School of Science and Biodiversity, Boulogne (France)	School group project designed by Chartier Dalix Architects [42]	Built in 2014.
Tower D2, Paris (France)	Office building design by Bechu & Associates [43]	Built in 2015
Biolum’Reef, Marseille (France)	Conceptual eco-green inhabited reef designed by Tangram Architects [44]	Unbuilt conceptual project
Spiral house, Paris (France)	Individual house design by In Situ Architecture [45]	Built in 2007
CIRC, Lyon (France)	Headquarters of the International Agency for Research on Cancer designed by Art & Build [23]	Under construction

**Table 2 biomimetics-07-00212-t002:** Examples of French academic and industrial research programs in biomimicry for the built environment carried out over the last ten years. (*) indicates the organization that initiates the program.

Research Programs for System Development	Description of the Project	Partnerships (* Project Leader)	Maturity
SymBiO_2_	5-year research program for the development of biological solar panels for microalgae culture [49,50,51]	XTU Architects * Lab Gepea, France	Prototype (Algo House pavilion, Paris) [51]
Biolum’Archi	3-year research program for the development of bioluminescence light system [52]	Tangram Lab * Tangram + Rougerie Mediterranean Institute of Oceanology	Research
R&D project on ‘biomimetic kinetic solar shading device’	Research for the development of an adaptive shading system inspired by flowers’ deployment mechanism [23,53]	Art & Build *	Prototype of the facade Pho’liage
Research programs for methodology development			
Bioinspired program, including a doctoral thesis	Applied research program for the development of bioinspired building systems [54]	Nobatek/INEF4 * I2M lab, MNHN (Mecadev)	PhD completed in 2022
Doctoral thesis	Applied research for the development of tools to design multifunctional building skins [35]	Ceebios * MNHN (Mecadev), Vicat	PhD completed in 2021
Doctoral thesis	Applied research for the development of tools to design regenerative urban environments [36]	Ceebios * MNHN (Cesco)	PhD completed in 2022
Research in biomimetics by the company Saint Gobain	Comparative review of living systems’ thermal regulation processes for building solutions [55]	Saint-Gobain * MNHN-Lab MECADEV	Journal paper published in 2019
BioMIG	Applied research on bioinspired materials for cross-industry sustainable applications [56]	Ceebios, consortium (industry + public & private research)	Multi-year program started in 2021

**Table 3 biomimetics-07-00212-t003:** Four calls for projects integrating biomimicry and regenerative requirements.

Projects	Description	Status
Marine Biomimetic Centre, Biarritz (France)	5800 m^2^, public building. Open innovation centre in biomimetics and marine biology [36]	2018—contract awarded by Patrick Arotecharen architecture studio
Theatre hall, Saintes (France)	120 m^2^, public building for the students of the high school of Saintes [64]	2019—unknown
Conservatory, Senlis (France)	2000 m^2^, renovation and extension of a military barrack into an academy of dance and music [65]	2022—evaluation of the building proposals
Business Park, La Base 217, Essonne (France)	35 hectares of a mixed-use business park [66]	2021—unknown

**Table 4 biomimetics-07-00212-t004:** Overview of the diversity of courses in biomimicry in architecture and urban planning taught by French institutions.

Short-Term Courses	Description of the Courses	Public	Sessions
ENSAG: National School of Architecture of Grenoble	25-h course in biomimicry for regenerative design in architecture and urban planning. Optional course	40 undergraduate students in architecture and design	1 session per year since 2018
ENSA PVS: National School of Architecture of Paris Val de Seine	35-h course in biomimicry and multi-regulation for the design of bio-inspired building envelopes. Optional course (in partnership with ICADE and Ceebios)	40 undergraduate students in architecture and civil engineering	1 session in 2018
ENSA PLV: National School of Architecture Paris la Villette	1-year course in digital architectural design and biomimicry. Optional course (in partnership with MAP MAACC CNRS)	15 to 25 undergraduate students in architecture	2 semesters, 1 day per week
Master’s in urban planning and development; University of Toulouse Jean-Jaurès	1-year course in new forms of innovation in urban planning through the biomimetic approach. Master course and research on nature-based solutions [74]	50 undergraduate students in urban planning	1 session per year since 2018
Long-term courses			
NID: Master of Sciences Nature Inspired Design, Paris	1-year master’s degree designed to train project managers in design and biomimicry, capable of mobilizing bio-inspired/nature-based solutions to contemporary challenges [75]	12 postgraduate students per year	1 session per year since 2020
BIM-Master of Bio-inspired Materials, Pau	2-year master’s exploring innovative approaches to develop materials inspired by living systems with special regard to their composition, function, structure, architectures, and processing, and in line with environmental sustainability [76]	12 postgraduate students or professionals with major education in biology, physics, and chemistry	1 session per year since 2022
Ceebios (Paris)	2-days training in bio-inspired architecture and urban planning for a professional of the building sector [31]	Professional	3 sessions per year since 2022

## Data Availability

Not applicable.

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
