# Peer review of "Has Biomimicry in Architecture Arrived in France? Diversity of Challenges and Opportunities for a Paradigm Shift"

_biomimetics, 2022, doi:10.3390/biomimetics7040212_

Round 1

Reviewer 1 Report

The paper tracks implementations of biomimicry inspired design in architecture in France. The article is very short for a journal contribution. The English language is often unclear and needs another round of editing by an experienced academic writer and native speaker. Correct tenses (past and present are not coordinated well) throughout.

The article does not set a clear scope in the introduction. The last paragraph in Section 1.1 is not clear. The writing style seems to follow an evocative design writing approach which unfortunately is not well suited to a journal paper that should be focused on making research outcomes and insights clear and explicit.

The number of cases discussed is entirely insufficient, as the article states already. What are the selection criteria to pick out these cases? I can see the contribution of an article that maps the TYPES and QUALITIES of biomimicry introduced via selected examples but this is not what the paper seems to aim to do.

Biomimetics seems to be presented as a default "better" approach compared to others. However biomimetics in itself does not guarantee any success or progress in the proposals designed through them. Why should policy makers prioritize biomimicry based design?

Overall I find though the observations presented are interesting and valuable for a specific audience they are not framed in a systematic way that allows others to use insights or knowledge offered in this paper. Could the contribution to existing research in this field be made more clear please?

The conclusion is entirely insufficient and gives an example of lack of content of the paper.

Author Response

Dear reviewer,

Please find attached the revised manuscript and responses to your comments.

My best regards, 

Estelle Cruz, Eduardo Blanco, Kalina Raskin, Fabienne Aujard

Reviewer 2 Report

1. Good work and interesting approach to this discipline.

2. I do not know why authors the authors do not want to relate the works of France with what happens in other countries. Only that comparison will make the article especially relevant.

3. In an article on architecture, I think it is essential that there be images, plans and/or infographics of the proposed spaces.

4. Very important in my opinion: Why are those examples relevant to biomimicry? What do each of them contribute? I haven't been able to find any information about it.

5. The same happens with the Urban Policies section.

6. Regarding the bibliography, I believe that many more articles are needed to relate what the authors propose with the reality of the building. I have one suggestion <https://www.sciencedirect.com/science/article/pii/S2095263518300773> but there could be many more.

7. The conclusions have to be improved because they are too general, what is their value compared to what has already been commented on in the abstract?

Author Response

(The authors gave the same response as above.)

Round 2

Reviewer 1 Report

The paper has significantly improved since the first version I reviewed, and is much more informative as a contribution to research as a result. Please take care of the many grammar and spelling mistakes still found throughout the paper before publishing. However this will not require another round of review comments.

Reviewer 2 Report

Thank you authors for take into account previous comments.

The article is good in its ambition, but it lacks in its scientific procedure.